# *p*-nitrobenzyloxycarbonyl protective group as key to automated glycan assembly of neutral human milk oligosaccharides

Mei-Huei Lin[1,2,4], Yan-Ting Kuo[1,2,3,4], Kim Le Mai Hoang[3] & Peter H. Seeberger [1,2] ✉

Human milk oligosaccharides (HMOs) have highly diverse and branched structures that present a significant challenge for chemical synthesis. Here we show that masking the amino group in glucosamine with a *p*-nitrobenzyloxycarbonyl (*p*NZ) protecting group enhances coupling and deprotection efficiency during automated glycan assembly (AGA) of homogeneous HMOs, including the lacto-*N*-tetraose (LNT), lacto-*N*-neotetraose (LNnT), lacto-*N*-fucopentaose (LNFP), lacto-*N*-difuco-hexaose (LNDFH), and lacto-N-neohexaose (LNnH) series. Deprotection strategies are developed to achieve excellent purity of linear and branched HMO structures. The end-to-end tractability of *p*NZ-protected oligosaccharides underscores the robustness of this approach, while three orthogonal deprotection pathways offer synthetic versatility for HMO compounds.

Breast milk is the primary source of nutrition for infants during their early development, supporting a wide range of physiological functions[1–3]. Human milk oligosaccharides (HMOs) are the third most abundant component of breast milk and are highly diverse in structure and function[4,5]. Comprehensive LC–MS surveys and databases have reported more than 200 HMOs, from which over 150 distinct structures were elucidated across both neutral and sialylated HMOs[6–9]. However, understanding the functional roles of HMOs remains limited due to difficulties in obtaining these complex glycans in pure form and sufficient quantities. Since isolating large amounts of pure HMOs from natural sources is challenging, synthetic approaches are essential to explore the biological potential of HMOs[10,11].

Various strategies have been developed to prepare HMOs, including chemical synthesis[12–14], enzymatic synthesis[15], chemo-enzymatic synthesis[16–18], microbial fermentation[19], and mammalian cell culture[20]. The backbone of HMOs, consisting of a lactose unit at the reducing end, extended by *N*-acetylglucosamine (GlcNAc) and galactose (Gal), is often decorated with fucose (Fuc) and/or *N*-acetylneuraminic acid (Neu5Ac). Notably, GlcNAc contains a C-2 amino group. The choice of protecting group at this position is crucial, as it can

significantly impact glycosylation reactivity[21]. Protecting groups that can be removed under mild and selective conditions offer a unique access point for fluorescent labeling[22], bio-orthogonal chemistry[23], and modulation of pharmacological properties of HMOs.

Automated glycan assembly (AGA) accelerates the synthetic process of a wide variety of complex oligosaccharides[24–29]. In solution-phase oligosaccharide synthesis, amine groups are commonly masked using trichloroethylchloroformate (Troc) and phthaloyl (Phth). These participating protecting groups promote the formation of 1,2-*trans* glycosides and suppress the formation of 1,2-oxazoline byproducts compared to natural acetyl (Ac) group. However, the Troc group is base-labile, making it incompatible with AGA protocols where concentrated basic solutions are introduced periodically. In contrast, Phth is base-stable and thus compatible with basic steps, but its removal typically relies on hydrazinolysis or related nucleophiles, which can demand large excess of reagents and, depending on substrate, elevated temperature or prolonged reaction times[30,31]. These conditions can compromise sensitive glycosidic bonds and other temporary protecting groups, making them not applicable to further modifications or global deprotection following AGA (Fig. 1a). Trichloroacetyl

[1]Department of Biomolecular Systems, Max-Planck Institute of Colloids and Interfaces, Potsdam, Germany. [2]Institute of Chemistry and Biochemistry, Freie Universität Berlin, Berlin, Germany. [3]GlycoUniverse GmbH & Co. KGaA, Potsdam, Germany. [4]These authors contributed equally: Mei-Huei Lin, Yan-Ting Kuo. ✉e-mail: peter.seeberger@mpikg.mpg.de

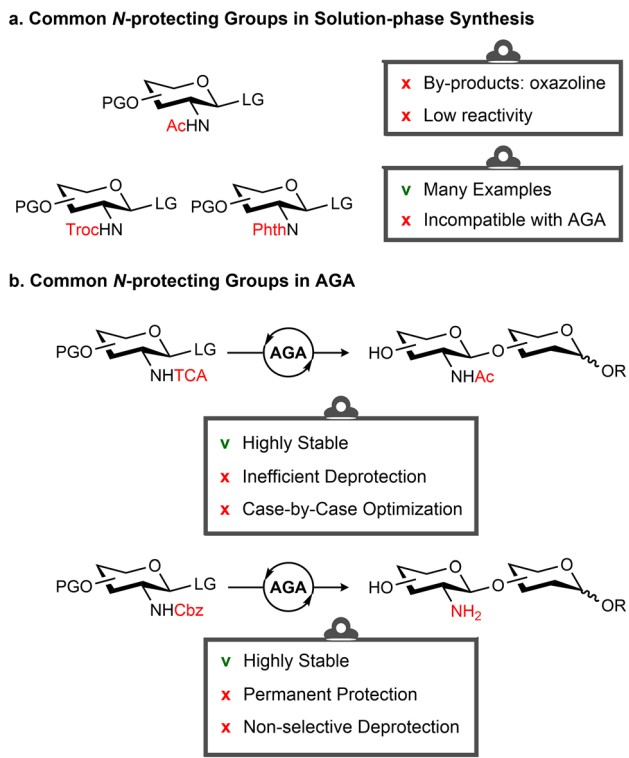

### a. Common *N*-protecting Groups in Solution-phase Synthesis

x By-products: oxazoline
x Low reactivity

v Many Examples
x Incompatible with AGA

### b. Common *N*-protecting Groups in AGA

v Highly Stable
x Inefficient Deprotection
x Case-by-Case Optimization

v Highly Stable
x Permanent Protection
x Non-selective Deprotection

### c. This Work: *p*-nitrobenzyloxycarbonyl (*p*NZ) in AGA

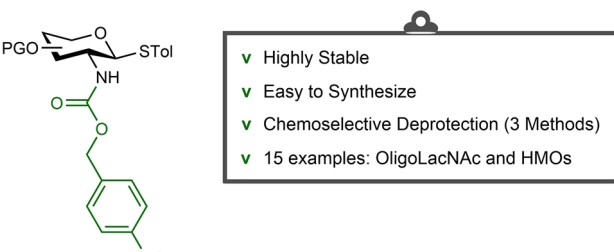

v Highly Stable
v Easy to Synthesize
v Chemoselective Deprotection (3 Methods)
v 15 examples: OligoLacNAc and HMOs

**Fig. 1 | *N*-Protecting groups for glycan assembly. a** Common groups for solution-phase synthesis; (**b**) Common groups for AGA; (**c**) This work: *p*NZ as *N*-protecting group in AGA. PG: protecting group, LG: leaving group (thioether or glycosyl phosphate).

(TCA) and carboxybenzyl (Cbz) groups were previously explored to overcome the challenges (Fig. 1b)[32,33]. The greater stability of TCA and Cbz groups often comes at a cost: multiple glycosylation cycles and the need to convert to a more reactive phosphate leaving group to overcome otherwise sluggish couplings. Simultaneous cleavage of multiple TCA and Cbz protecting groups often requires extended hydrogenolysis and can suffer from selectivity issues, particularly when Cbz is also present on linkers, leading to parallel unmasking of multiple amines and complicating selective *N*-acetylation. Prior experiments reported incomplete TCA reduction, fucose cleavage, and solubility problems[33,34].

Here, we aimed to develop reliable syntheses of a collection of diverse neutral HMOs using AGA on a commercial Glyconeer® synthesizer as well as home-built instruments. A key aspect of this development was the introduction of the *p*-nitrobenzyloxycarbonyl (*p*NZ)[35-37], a carbamate-type protecting group. Initially, *p*NZ was employed in peptide chemistry for the protection of amino side chains, particularly to mask the amine functionalities of several amino acids. In these solution-phase studies[36,38], *p*NZ-protected donors exhibit lower reactivity compared to other amino-protected donors, necessitating higher temperatures and longer reaction times.

However, in solid phase synthesis, this moderate reactivity helps suppress donor decomposition and side reactions, thereby improving synthetic efficiency[24,39,40]. *p*NZ-protected glucosamine building blocks displayed excellent stability and yielded exclusively 1,2-*trans* glycosides. In addition, these shelf-stable thioglycosides can be used directly without the need to convert them into glycosyl phosphates (Fig. 1c). The preparation of these building blocks was straightforward and efficient. In this work, *p*NZ was key to the assembly of branched and linear HMO structures through AGA. We also demonstrated that *p*NZ is a very versatile and orthogonal protecting group, which can be selectively removed in three different ways.

## Results and discussion

### Preparation of *N-p*NZ protected building blocks

To accommodate the diverse native linkages found in HMOs, we designed four *N-p*NZ-protected glucosamine building blocks optimized for AGA. These building blocks feature a permanent benzyl ether (Bn) protecting group as well as two orthogonal temporary protecting groups: fluorenylmethoxycarbonyl (Fmoc) and levulinoyl (Lev). The glucosamine set includes 4-O-Fmoc (**3**), 4-O-Lev (**4**), 3-O-Fmoc (**6**), and 3-O-Lev-4-O-Fmoc (**8**) (Fig. 2). In brief: thioglycoside **1** containing a C-2 *p*NZ group was synthesized from D-glucosamine in four steps[36]. Next, one-pot silylation and benzylation were employed: Addition of 1,1,1,3,3,3-hexamethyldisilazane (HMDS) and trimethylsilyl trifluoromethanesulfonate (TMSOTf) in anhydrous acetonitrile (MeCN) afforded per-O-silylation of **1**[41]. Subsequently, 4,6-O-benzylidenation and 3-O-benzylation by addition of benzaldehyde (PhCHO), triethylsilane (Et₃SiH) and trifluoromethanesulfonic acid (TfOH)[42], followed by regioselective ring opening of the 4,6-O-benzylidene group with trifluoroacetic acid (TFA) and triethylsilane (Et₃SiH) gave intermediate **2**. 4-O-Fmoc glucosamine donor **3** and 4-O-Lev glucosamine donor **4** were subsequently prepared from **2**.

3-O-Fmoc glucosamine donor **6** was prepared from **1** by 4,6-O-benzylidenation with benzaldehyde dimethyl acetal (PhCH(OMe)₂) and a catalytic amount of camphor sulfonic acid (CSA) in MeCN/THF, followed by introduction of Fmoc group at the C-3 position. The intermediate was subjected to reductive opening of the 4,6-O-benzylidene ring using a combination of dichlorophenylborane (BPhCl₂) and TMSOTf, yielding intermediate **5**. Since Fmoc is a base-labile protecting group, acidic benzylation was carried out to furnish **6**[41].

3-O-Lev-4-O-Fmoc glucosamine donor **8**, bearing two temporary protecting groups at the C-3 and C-4 positions, was subsequently prepared. The Lev group was introduced at C-3 following 4,6-O-benzylidene protection. Selective reductive ring opening and introducing Fmoc at C-4 afforded building block **8** in excellent yield.

All synthetic intermediates were readily isolated by simply washing under hot ethanol (*ca.* 60 °C) as excess reagents and by products were completely soluble, whereas the desired products precipitated and were filtered from the crude solution before a final flash column chromatography.

### AGA using *N-p*NZ protected glucosamine building blocks

The diLacNAc tetramer, Galβ1-4GlcNAcβ1-3Galβ1-4GlcNAcβ **10**, served as target molecule for optimizing AGA conditions. Generally, glycosylation efficiency and stereoselective outcomes are influenced by many factors, including concentration, temperature, protecting groups, and solvent[40,43-48]. For AGA, the reaction vessel was loaded with a set amount of modified polystyrene resin carrying a photolabile linker. To prepare the resin for the first glycosylation, an acidic wash was followed by glycosylation on solid support. The instrument delivered a solution of building block and activator solution, under an inert atmosphere and precise temperature. Trace amounts of unreacted hydroxyl groups were capped via acidic acetylation, followed by orthogonal removal of a temporary protecting group at the

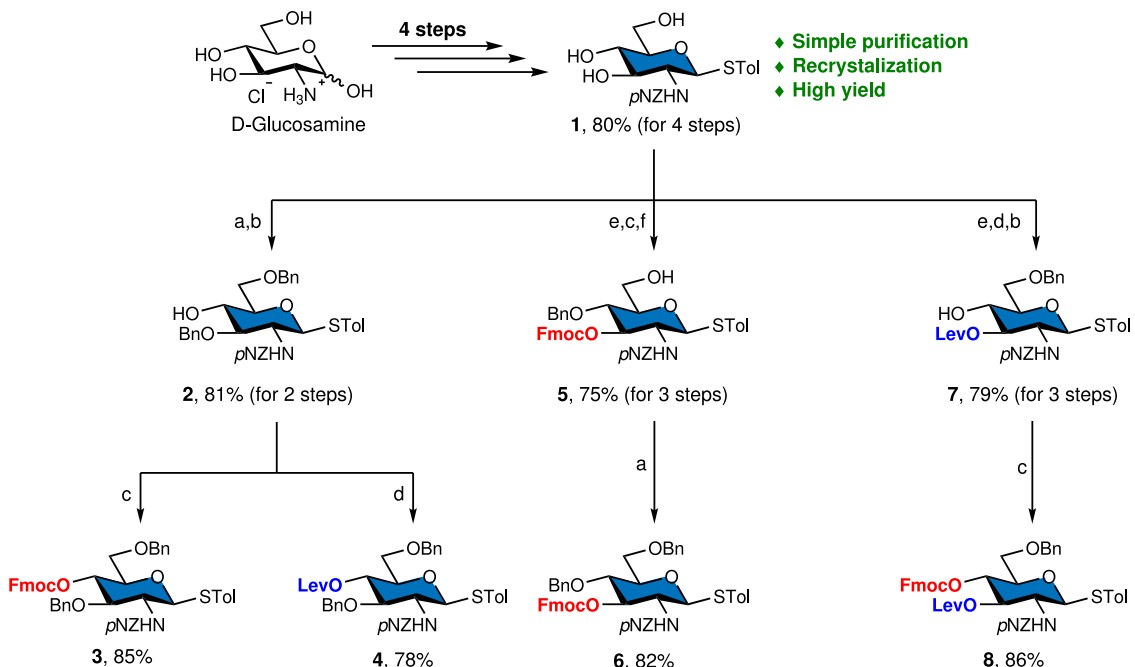

**Fig. 2 | Synthesis of the N-pNZ protected glucosamine building blocks 3, 4, 6 and 8. a** 1 HMDS, TMSOTf, MeCN, r.t. 2) PhCHO, Et$_3$SiH, TfOH, CH$_2$Cl$_2$, 0 °C to r.t.; **b** TFA, TFAA, Et$_3$SiH, CH$_2$Cl$_2$, 0 °C to r.t.; **c** FmocCl, pyridine, CH$_2$Cl$_2$, r.t.; **d** EDC, DMAP, LevOH, CH$_2$Cl$_2$, 0 °C to r.t.; **e** PhCH(OMe)$_2$, CSA, MeCN/THF r.t.; **f** PhBCl$_2$, Et$_3$SiH, CH$_2$Cl$_2$, −78 °C.

designated elongation site. The cycle was programmed to repeat fully autonomously until the desired glycan structure was built[49].

We recently identified optimal initiation and coupling temperatures for 3-O-Fmoc galactoside **9**[40,50]. The initiation temperature ($T_1$) was set at −40 °C for five minutes, followed by coupling temperature ($T_2$) at −20 °C for 20 min. These conditions were kept unchanged, while efforts focused on optimizing the conditions for 4-O-Fmoc glucosamine donor **3**. After 30 min incubation at −20 °C, most of **3** remained unactivated. Gratifyingly, at −10 °C this glycosyl donor was fully activated (Fig. 3a, b and Supplementary Fig. 1). Following photocleavage and normal-phase HPLC purification, tetrasaccharide **10** was isolated in 27% overall yield. The analytical HPLC of the crude sample revealed traces of deletion sequences. A further increase in the glycosylation temperature to 0 °C resulted in accelerated decomposition of **3** and significantly reduced the yield of the target compound **10**. Thus, the coupling temperature for **3** was set at −10 °C for 30 min.

To address the traces of deletion sequences observed, we examined the concentrations of TfOH and TMSOTf, as their acidity can critically affect glycosidic bond activation and stability. For TfOH, an optimal threshold concentration of 0.15 equivalents per glycosyl donor was observed. The effect of TMSOTf concentration in the acidic wash module was also examined (Fig. 3c, d). Under concentration of 1% v/v TMSOTf, signs of glycosidic bond cleavage was observed. Reducing TMSOTf concentration tenfold to 0.1% v/v completely suppressed cleavage, raised isolated yield to 37% as observed by crude HPLC analysis. Next, the solvent ratio and an extra basic wash with 10% pyridine solution were examined, but these had only minor impact on the glycosylation efficiency (Fig. 3e, f and Supplementary Fig. 2).

## Automated synthesis of pNZ-protected oligo-LacNAc & HMOs (Fig. 4)

With optimized coupling conditions in hand, the substrate scope of HMOs was explored. The synthesis of HMOs fragments containing LacNAc type-2 linkage (Galβ1-4GlcNAcβ1-3) yielded tetramer **10** (37%), hexamer **18** (26%), and octamer **19** (20%) using 3-O-Fmoc galactoside **9** and 4-O-Fmoc glucosamine donor **3**.

Oligo-LacNAc and HMOs differ in the first monosaccharide at the reducing end: while oligo-LacNAc begins with glucosamine, HMOs contain glucose. 4-O-Fmoc glucoside **11**, from our building block repertoire[32], was coupled at an initiation temperature ($T_1$) of -20 °C for five minutes, followed by a coupling temperature ($T_2$) at 0 °C for 20 min. Removal of Fmoc protecting group exposed a C-4 hydroxyl group for glycosylation using building blocks **9** and **3** to furnish linear tetrasaccharide **20** (36%) and hexasaccharide **21** (28%).

Oligosaccharides containing the LacNAc type-1 epitope (Galβ1-3GlcNAc), are abundant in HMOs and associated with a variety of developmental and pathological conditions[51]. To construct the linear Galβ1-3GlcNAc **22**, 3-Fmoc glucosamine donor **6** was coupled at 0 °C for 40 min to overcome its lower reactivity to give tetrasaccharide **22** in 32% yield.

Many neutral HMOs do contain fucose as a crucial epitope[33]. The Fucα1-3GlcNAc branch was introduced into the linear LacNAc type-2 tetramer, yielding pentasaccharide **23**. A selective deprotection strategy was implemented to install both Galβ1-4GlcNAc and Fucα1-3GlcNAc linkages. First, the Lev-protecting group on the terminal glucosamine was selectively removed using a solution of hydrazine acetate, exposing a hydroxyl group for the attachment of per-benzyl fucoside **13**. Thereafter, the 4-O-Fmoc group of glucosamine was removed with piperidine, revealing a second hydroxyl group for glycosylation with 3-O-Fmoc galactoside **9**. This stepwise strategy furnished pentasaccharide **23** in 29% isolated yield. The same strategy for deprotection of Fmoc and Lev groups was applied in the synthesis of LacNAc type-1 pentasaccharide **24**, to introduce the Fucα1-4GlcNAc branch. In order to prevent complications during Fmoc deprotection, the 2,3-O-di-benzoyl galactoside **14** was introduced as the terminal unit. Building block **14** required a higher activation temperature and the coupling was performed with $T_1$ at -20 °C for five minutes, followed by $T_2$ at 0 °C for 20 min to yield 21% of pentasaccharide **24**.

For symmetrically branched HMOs, the 6-O-Lev-3-O-Fmoc galactoside **15** provided the appropriate branching points to construct both the GlcNAcβ1-6Gal and GlcNAcβ1-3Gal linkages. However, the presence of an additional electron-withdrawing group on building block

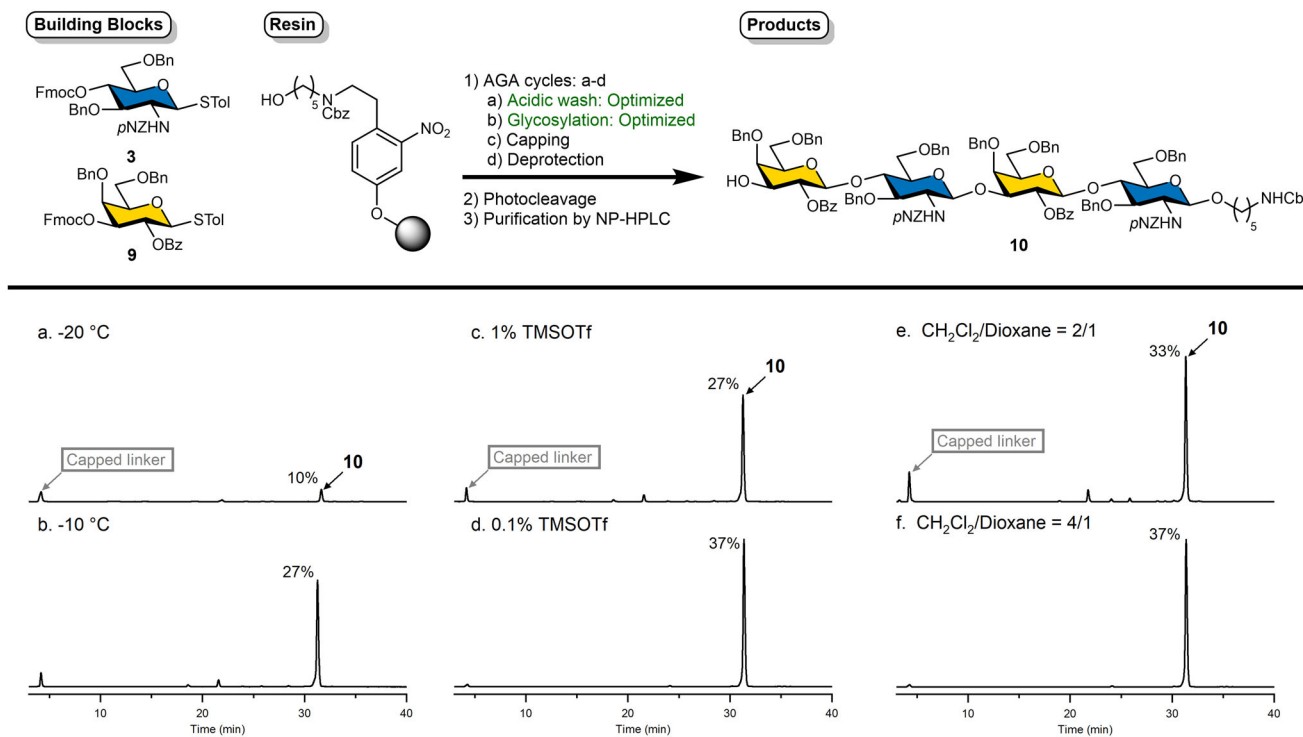

**Fig. 3 | AGA optimization using analysis of crude HPLC spectra. a** Glycosylation at −20 °C; **b** Glycosylation at −10 °C; **c** 1% TMSOTf as acidic wash solution; **d** 0.1% TMSOTf as acidic wash solution; **e** 2/1 ratio of CH₂Cl₂/Dioxane; **f** 4/1 ratio of CH₂Cl₂/Dioxane.

**15** necessitated a higher activation temperature when compared to 3-O-Fmoc galactoside **9**[50]. During the synthesis of hexasaccharide **25a**, the 6-O-Lev and 3-O-Fmoc groups on **15** were sequentially removed, exposing two hydroxyl groups. Bis-glycosylation using 4-Fmoc glucosamine donor **3** was performed at both the 6-OH and 3-OH positions of the galactose unit. A subsequent bis-glycosylation step with 3-O-Fmoc galactoside **9** produced branched pentasaccharide **25a** in 29% isolated yield. For comparison, the N-pNZ-protected glucosamine donor **3** was replaced by N-TCA-protected **17a** and N-Cbz **17b** glucosamine donors to obtain hexasaccharides **25b** and **25c**, respectively. The pNZ-protected donor afforded a higher isolated yield (29%) over multiple glycosylation cycles, compared to the TCA-protected (16%) and the Cbz-protected donor (6%). In the case of the TCA-protected donor, the absence of prominent deletion sequences suggested that glycosidic linkages may undergo acid-catalyzed hydrolysis during the synthesis[52]. In contrast, the use of the Cbz-protected donor resulted in the deletion of sequences due to poorer glycosylation efficiency (Supplementary Fig. 3). Another symmetrically branched HMO containing the Fucα1-3GlcNAc motif was examined. The efficiency of the fucosylation and the final galactosylation was slightly reduced, likely due to steric effect. By increasing the equivalents of building blocks **9** and **13** from 6.5 to 10, target octasaccharide **26** was obtained in 19% isolated yield. After coupling, the hydrolyzed building blocks were recovered by rerouting the reaction mixture to the fraction collector.

To synthesize unsymmetrically branched decasaccharide HMO **27**, we introduced 4-O-Lev glucosamine donor **4** to facilitate a more complex synthetic pathway. This building block was activated similar to 4-O-Fmoc glucosamine donor **3**, as different ester groups did not markedly impact the activation temperature[50]. The architecture of decasaccharide **27** featured two distinct branching points on two identical galactose terminals. Initially, the 6-O-Lev group on the first galactose was removed to expose the C-6 hydroxyl group that was then glycosylated with 4-O-Lev glucosamine donor **4**. Next, the 4-O-Lev on this glucosamine was removed, and the resulting hydroxyl group was used to react with 6-O-Lev-3-O-Fmoc

galactoside **15**. The resulting tetrasaccharide contained two Fmoc and one Lev groups that were promptly cleaved to reveal three hydroxyl groups. A tris-glycosylation step with 4-O-Fmoc glucosamine donor **3** was followed by another tris-glycosylation with galactose building block **9**, completing the synthesis of decasaccharide **27** in 16% isolated yield. The automation sequence for the unsymmetrically branched heptasaccharide **28** combined elements of the sequences used to synthesize glycans **23** and **27**. The 6-O-Lev on galactose was removed, followed by glycosylation with 4-O-Lev glucosamine donor **4** and 2,3-O-di-benzoyl galactoside **14**. Subsequently, the 3-O-Fmoc on galactose **15** was removed to expose the hydroxyl group. The Lewis-X epitope, Galβ1-4(Fucα1-3)GlcNAc, was introduced using the protocol applied for pentasaccharide **23**, yielding 17% of heptasaccharide **28**. Hexasaccharide **29** contains two fucose branches at glucose and glucosamine. Initially, 3-O-Lev, 4-O-Fmoc glucoside **12** was introduced to construct the Fucα1-3Glc and Galβ1-4Glc linkage. The Fmoc group of galactose was removed to expose the C-3 hydroxyl group that was then glycosylated with 3-O-Lev-4-O-Fmoc glucosamine donor **8**. However, due to steric hindrance, the reactivity of the C-3 hydroxyl group in galactose diminished, necessitating an increase in coupling time from 30 to 45 min. The Lewis-A epitope Galβ1-3(Fucα1-4)GlcNAc was added using the automation protocol employed for the synthesis of pentasaccharide **24** to yield 17% of hexasaccharide **29**.

α1,2-Fucosylated glycans represent a distinctive subset of HMOs, characterized by the presence of α1,2-linked fucose residue, also commonly associated with the H-antigen[5]. These structures are predominantly found in the milk of so-called secretor mothers, whose active *FUT2* gene enables the enzymatic addition of fucose in an α1,2 linkage to precursor glycans. Owing to their structural uniqueness and biological relevance, α1,2-fucosylated HMOs are believed to play critical roles in shaping the infant gut microbiota and providing protection against pathogenic infections[53]. 2-Fmoc galactoside **16** was used to synthesize pentasaccharide **30** (21%) and hexasaccharide **31** (15%) yield.

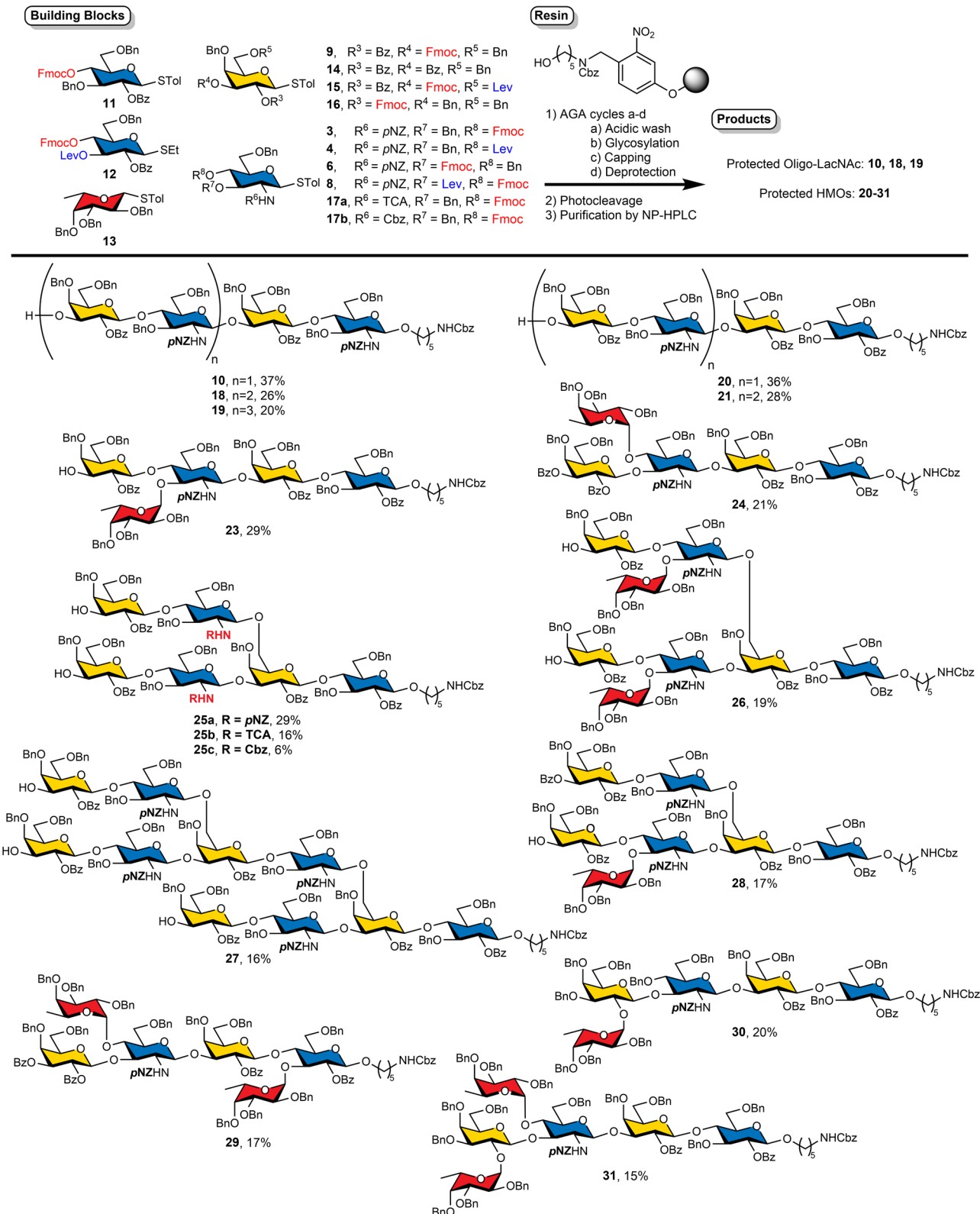

**Fig. 4 | Automated synthesis of protected oligo-LacNAc and HMOs. Overall yields indicated for each product.**

## Modification of protected oligo-LacNAc and HMOs

The pNZ group was removed by reduction of the nitro group in a two-step protocol. pNZ was first reduced to the p-aminobenzyloxycarbonyl derivative that underwent a spontaneous 1,6-electron pair shift, afforded quinonimine methide and carbamic acid. The transient carbamic acid subsequently decomposed to release the free amine[35]. The transformation was accelerated in the presence of an acid catalyst[35]. Three reducing conditions including the commonly used Tin(II) chloride (SnCl₂)[35], Zinc–copper couple (Zn/Cu)[54], and the recently reported tetrahydroxydiboron (B₂(OH)₄)/4,4′-bipyridine[55] were examined. All conditions successfully reduced the pNZ group to the free amine with moderate to high yield (70–89%). The boronic acid method

**a. Deprotection Condition:**

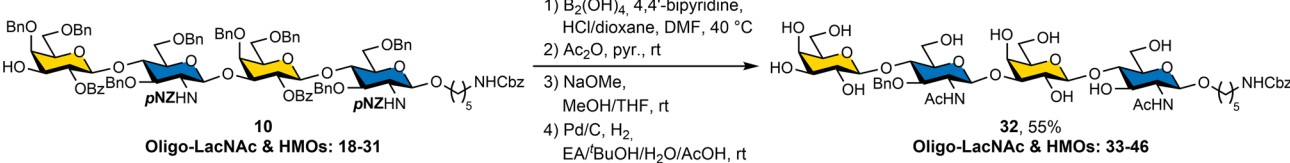

**10**
Oligo-LacNAc & HMOs: 18-31

1) B$_2$(OH)$_4$, 4,4'-bipyridine, HCl/dioxane, DMF, 40 °C
2) Ac$_2$O, pyr., rt
3) NaOMe, MeOH/THF, rt
4) Pd/C, H$_2$, EA/$^t$BuOH/H$_2$O/AcOH, rt

**32**, 55%
Oligo-LacNAc & HMOs: 33-46

**b. Collection of LacNAc-based oligosaccharides and HMOs**

**32**, 55%

**33**, 50%

**34**, 43%

**35**, LNnT, 55%

**36**, pLNnH, 52%

**37**, LNT, 57%

**38**, LNFP III, 55%

**39**, LNFPII, 50%

**40**, LNnH, 45%

**41**, DFLNnH, 42%

**42**, iLND, 40%

**43**, FLNnH II, 45%

**44**, LNDFH II, 40%

**45**, LNFP I, 42%

**46**, LNDFH I, 35%

Linker = $\sim\!\!O\!\!\sim\!\!\sim\!\!\sim\!\!NH_2$

● Glc   ● Gal   ▲ Fuc   ■ GlcNAc

**Fig. 5 | Overview of oligosaccharide products and deprotection conditions. a** HMOs deprotection conditions; **b** Collection of neutral linear, symmetrical- and unsymmetrical-branched LacNAc-based oligosaccharides and HMOs.

was chosen for pNZ deprotection, owing to its overall efficiency, mildness, and the avoidance of toxic heavy metals. In the second step, N-acetylation was conducted to further modify the amino group of glucosamine residues in oligo-LacNAc and HMO structures, resulting in the formation of N-acetylglucosamine within these structures. Then, a solution of sodium methoxide in MeOH was used to remove all ester-protecting groups, followed by hydrogenation with Pd/C catalyst to remove all remaining protecting groups, including benzyl ethers and benzyloxycarbonyl (Cbz), to afford the desired HMOs (Fig. 5a).

Following reverse-phase HPLC purification, oligo-LacNAc derivatives **32-34** were isolated in overall yields ranging from 43% to 55%

(Fig. 5b). The linear HMOs LNnT **35** and pLNnH **36** were synthesized in 55% and 52% yield, respectively. The tetrameric HMO LNT **37**, incorporating a Galβ1-3GlcNAc linkage, was obtained in 57% yield. Minor deletion sequences lacking fucose residues were detected after the global deprotection step, although strongly diminished when compared to syntheses involving building blocks protected with N-TCA groups[33]. Linear mono-fucosylated HMOs, including LNFP III **38** and LNFP II **39**, were produced in yields comparable to their non-fucosylated counterparts (ca. 50%). Branched structures, including symmetrical LNnH **40**, DFLNnH **41**, and unsymmetrically branched iLND **42**, FLNnH II **43** were synthesized efficiently by routine

adaptation of the automation sequence. Finally, di-fucosylated HMOs, including LNDFH II **44**, bearing Fucα1-3 and Fucα1-4 epitopes, along with LNFP I **45** and LNDFH I **46**, featuring Fucα1-2 motifs, were successfully obtained. The reducing ends of the oligo-LacNAc and HMOs were covalently linked to a five-carbon aliphatic spacer bearing a terminal amine released from the photocleavable solid support for immediate conjugation on glass surfaces for glycan array fabrication or to various biomolecules of interest, such as carrier proteins for vaccine development or adjuvants. The pNZ protecting group can be selectively removed, providing non-acetylated amino glycans, for example chitooligosaccharides, that serve as a versatile handle for further transformations including bioconjugation or incorporation into natural products[32].

In conclusion, the p-nitrobenzyloxycarbonyl group was examined as a protecting group for the C-2 amine in glucosamine during automated glycan assembly. Glucosamine building blocks carrying pNZ protection enabled the efficient automated synthesis of human milk oligosaccharides and oligo-LacNAc structures. The new building blocks were prepared conveniently using a simple purification method. Investigations into temperatures, concentrations, and automation sequences led to a robust routine protocol for these challenging and manually labor-intensive target molecules. The high end-to-end efficiency of pNZ protection facilitated the synthesis of linear, branched, structurally diverse, and complex HMO variants using the Glyconeer® automated synthesizer. Moreover, three methods for the selective removal of the pNZ group were applied to enable product customization.

## Methods

### General
The synthesis and characterization of monomer building blocks (**3, 4, 6** and **8**), along with the synthetic protocols and HPLC chromatograms for AGA of protected-oligosaccharides (**10** and **18**–**31**) and deprotected-oligosaccharides (**32**–**46**), are provided in Supplementary Information. $^1$H, $^{13}$C and two-dimensional (2D) NMR spectra of the compounds described in the article are available in Supplementary Information.

### AGA
The automated synthesizer executes a series of commands that are combined into modules to achieve specific transformations (see Supplementary Information)

### Acidic wash module
Once the temperature of the reaction vessel has adjusted to the desired temperature of −20 °C by the cooling device, 1 mL of the Acidic Wash Solution is delivered to the reaction vessel. After three minutes, the solution is drained. Finally, the resin is washed with 3 mL $CH_2Cl_2$ (bubbling = 15 s) and drained.

### Glycosylation using thioglycosides module
Upon draining the $CH_2Cl_2$ in the reaction vessel, 1 mL of Building Block Solution containing the appropriate building block is delivered from the building block storing component to the reaction vessel through. After the temperature reaches the desired temperature ($T_1$), Activator Solution (1 mL) is delivered to the reaction vessel from the respective activator storing component to the reaction vessel. The glycosylation mixture is incubated for the selected duration ($t_1$) at the desired $T_1$, then the reaction temperature is linearly ramped to $T_2$. Once $T_2$ is reached, it is maintained and the reaction mixture is incubated for an additional time ($t_2$). Once the incubation time is finished, the reaction mixture is drained and the resin is washed with $CH_2Cl_2$ (1 × 2 mL for 15 s), then dioxane (1 × 2 mL for 15 s), and finally $CH_2Cl_2$ (2 × 2 mL for 15 s).

### Pyridine wash module
The resin is washed with DMF (2 × 3 mL for 15 s). Then Pre-capping Solution (2 mL) is delivered into the reaction vessel and incubated for 3 min. The resin is then washed with $CH_2Cl_2$ (3 × 2 mL for 15 s).

### Capping module
The resin is washed with DMF (2 × 3 mL for 15 s). Then Pre-capping Solution (2 mL) is delivered into the reaction vessel and incubated for 3 min. The resin is then washed with $CH_2Cl_2$ (3 × 2 mL for 15 s). Upon washing, Capping Solution (4 mL) is delivered and the temperature is adjusted and maintained 25 °C. The resin and the reagents are incubated for 20 min. The solution is then drained from the reactor vessel and the resin is washed with $CH_2Cl_2$ (3 × 3 mL for 15 s).

### Fmoc deprotection module
The resin is first washed with DMF (3 × 3 mL for 15 s), and then Fmoc Deprotection Solution (2 mL) is delivered to the reaction vessel. After 5 min the reaction solution is drained and the resin is washed with DMF (3 × 3 mL for 15 s) and $CH_2Cl_2$ (3 × 3 mL for 15 s). After this module the resin is ready for the next glycosylation cycle.

### Lev deprotection module
The resin was washed with DMF (3 × 30 s) and DCM (1.3 mL) added to the reaction vessel. Lev Deprotection Solution (0.8 mL) was added to the reaction vessel, and the temperature was adjusted to 25 °C. After 30 min, the reaction solution was drained and the entire cycle was repeated twice more. After Lev deprotection was completed, the resin was washed with DMF, THF and DCM.

### Cleavage and normal phase purification
**Resin cleavage.** After automated synthesis, the resin was removed from the reaction vessel, suspended in $CH_2Cl_2$ (20 mL), and photocleaved in a continuous-flow photoreactor. A Vapourtec E-Series easy-MedChem, equipped with a UV-150 Photochemical reactor having a UV-150 Medium-Pressure Mercury Lamp (arc length 27.9 cm, 450 W) surrounded by a long-pass UV filter (Pyrex, 50% transmittance at 305 nm) was used. A Pump 11 Elite Series (Harvard Apparatus syringe pump at a flow rate of 1.0 mL/min was used to pump the mixture through a FEP tubing (i.d. 3.0 inch, volume: 12 mL) at 20 °C. The reactor was washed with 20 mL $CH_2Cl_2$ at a flow rate of 2.0 mL/min. The output solution was filtered to remove the resin and the solvent was evaporated in vacuo.

**Analytical NP-HPLC.** The crude product was dissolved in 4 mL of ethyl acetate (EtOAc) and analyzed using analytical HPLC (Agilent 1200 Series system). A YMC-Diol-300-NP column (150 mm × 4.600 mm I.D.) was used with a flow rate of 1.00 mL/min.

**Preparative NP-HPLC.** The crude product was dissolved in a 1:2 mixture of hexane and EtOAc and conducted on an Agilent 1200 Series system. A YMC-Diol-300-NP column (150 mm × 20 mm I.D.) was used with a flow rate of 15.00 mL/min.

### General deprotection procedure
Protected glycan, $B_2(OH)_4$ (22 mg), and 4,4'-bipyridine (0.1 mg) were all added to a sample vial. Then, DMF (3 mL) was added to a reaction mixture. Over 10 min, the reaction mixture changed color from transparent to purple and subsequently to yellow. After 15 min, 1 M of HCl in dioxane (50 μL) was added, and the reaction mixture was stirred at 40 °C for overnight. After the reaction, the reaction mixture was diluted with water and ethyl acetate. The organic layer was collected and washed with brine and evaporated in vacuo. Then, the crude compound was dissolved in $Ac_2O$/pyridine (4 mL, 1:1) and stirred for 3 h. Upon the completion of acetylation, the solvent was evaporated in

*vacuo*. The acetylated crude compound was dissolved in THF (5 mL) and sodium methoxide (0.5 M solution in MeOH, 0.5 mL) was added. The mixture was stirred at room temperature for two hours. Amberlite IR-120 (H⁺ form) was then added to quench. After neutralization, the reaction mixture was filtered and the solvent was removed *in vacuo*. The crude compound was used for hydrogenolysis without further purification. The crude compound from methanolysis was dissolved in a mixture of EtOAc:ᵗBuOH:H₂O (2:1:1, 4 mL) with AcOH (0.1 mL). Pd/C (10%) was added, and the solution was purged with N₂ and H₂ for ten minutes. The suspension was stirred under H₂ balloon for overnight. The insoluble material was removed by a CHROMAFIL ®Xtra, RC 0.45 syringe filter. The solid was washed once with MeOH and several times with water. The filtrate was collected and concentrated *in vacuo*.

### Reverse-phase purification
**Analytical RP-HPLC.** The crude product was dissolved in 3 mL of H₂O and analyzed using analytical HPLC (Agilent 1200 Series system). A Hypercarb column (150 mm × 4.6 mm I.D.) was used with a flow rate of 0.7 mL/min.

**Preparative NP-HPLC.** The crude product was dissolved in H₂O and conducted on an Agilent 1200 Series system. A Hypercarb column (150 mm × 10 mm I.D.) was used with a flow rate of 3.0 mL/min.

## Data availability
The raw data used to generate the figures and tables in this manuscript are available from the corresponding author on request.

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

## Acknowledgements
We gratefully acknowledge the Max-Planck Society for its generous financial support. Y.-T.K. and K.L.M.H. acknowledge financial support from the European Union's Horizon 2020 research and innovation program under the Marie Skłodowska-Curie grant agreement No 956758.

## Author contributions
K.L.M.H. and P.H.S. conceived and supervised the experiments and contributed to manuscript preparation. M.-H.L. and Y.-T.K. performed the synthetic work, conducted data analysis, and prepared the initial draft of the manuscript.

## Funding

## Competing interests
M.-H.L. and Y.-T.K. declare no competing interests. K.L.M.H. and P.H.S. declare a competing interest in GlycoUniverse GmbH & Co. KGaA, the company that commercializes the Glyconeer® automated synthesizer and glycan products, including all building blocks used in this study.
