## [Transparent Peer Review file · Nature Communications]

p-Nitrobenzyloxycarbonyl Protective Group as Key to Automated Glycan Assembly of Neutral Human Milk Oligosaccharides

Corresponding Author: Professor Peter Seeberger

Version 0:

Reviewer comments:

Reviewer #1

(Remarks to the Author)

In the manuscript by Seeberger and co-workers, they report solid phase synthesis of human milk oligosaccharides (HMOs) via automated glycan assembly. The processes rely on the stereoselective glycosylations mediated by the p-nitrobenzyloxycarbonyl group on the glucosamine moiety. A comparison of the oligosaccharide assemblies via pNZ-protected and TCA-protected glucosamine building blocks demonstrates the better efficiency of the former with respect to the latter (Query 1). Careful orthogonal protective selections and installations on the building blocks ensure the facile installations of different sugar moieties regioselectively via site-specific deprotections. Optimization of the conditions for glycosylation, washing, capping and deprotection enables the synthesis of a library of 15 different linear, symmetrical- and unsymmetrical-branched HMOs in acceptable yields. I recommend that the manuscript be accepted for publication after the following issues are addressed.

Queries:

1. A similar comparative study of the process may also be conducted with the carboxybenzyl (Cbz) protecting group on the glucosamine and reported in the SI.
2. I recommend that the words asymmetrical and asymmetrically be replaced with unsymmetrical and unsymmetrically.

Somnath Yadav

Reviewer #2

(Remarks to the Author)

This work describes an automated synthesis of a series of human milk oligosaccharides (HMOs), and the authors have achieved the rapid assembly of fifteen linear and branched HMO structures. Considering the potential applications of HMOs in drug development and the efficiency of the synthetic approach, this study undoubtedly possesses significant innovation and practical value. The authors emphasize that the use of p-nitrobenzyloxycarbonyl (pNZ) group as protective group of amino function was critical to achieving efficient synthesis of the target molecules. However, the reviewer does not share this view.

The authors devoted substantial space to detailing the synthetic procedures, including the preparation of various synthetic building blocks and the optimization of glycosylation reaction conditions. However, these descriptions fail to clearly demonstrate the key role played by pNZ group. While it is reasonable to believe that the pNZ protection may have contributed to improved synthetic efficiency—such as in the comparison of yields between hexasaccharide 25a and 25b, where the yield of 25a was enhanced by the use of pNZ—it remains unclear from the manuscript whether this was truly a determining factor for the success of the reactions.

From this perspective, the current work does not meet the publication standards of Nature Communications (NC). It is recommended that the authors consider submitting their work to more specialized chemical journals.

Reviewer #3

(Remarks to the Author)

The manuscript by Seeberger and colleagues describes the utilization of a p-nitrobenzyloxycarbonyl (pNZ) protecting group at C2-amino position of glucosamine to enable the efficient assemble of HMOs via automated glycan assembly. The authors first synthesized various glycosyl donors having C2-pNZ protecting groups and optimized their glycosylation preformation on a commercial Glycoeer® synthesizer. Later, they successfully synthesized various LacNAc derivatives along linear and branched HMOs. Finally, the authors demonstrated efficient and selective removal of pNZ group from assembled oligosaccharide, followed by N-acetylation and global deprotection to afford the desired HMOs in high yield and purity. I believe manuscript will appeal to a broad readership and represent a valuable contribution to Nature Communications, so I recommend its publication. The manuscript has a few minor concerns which can be improved to enhance its impact.

Comments and suggestions are as follows:

1. On page 1 left column, reference 6 does not appear to be the appropriate source to support the corresponding statement. Moreover, reference 6 reports that more than 150 HMOs with well-defined structures have been identified, whereas the authors state that "over 100 of those glycans have been fully characterized" which seems inconsistent with the cited source. Additionally, Reference 7 barely talks about HMOs and is not sufficient to support claims regarding their structural elucidation. The authors are advised to cite more relevant and recent reviews or primary research articles that comprehensively address the structural characterization of HMOs.
2. The following statement requires appropriate citation: Page 1, left column, "However, understanding the functional roles of HMOs remains limited due to difficulties in obtaining these complex glycans in pure form and sufficient quantities. Since isolating large amounts of pure HMOs from natural sources is challenging, synthetic approaches are essential to explore the biological potential of HMOs."
3. The following statement on Page 1, left column "Automated glycan assembly (AGA) accelerates the synthetic process of a wide variety of complex oligosaccharides" requires appropriate and balanced citation. While the authors reference three of their own publications, it is important to acknowledge that multiple research groups have developed automated glycan synthesis platforms. Please include additional citations to relevant work from other groups to provide a more comprehensive and unbiased overview of advancements in this area.
4. The following statement on Page 1, left column "However, NHTroc and NPhth are base-labile and require rather harsh deprotection conditions, such as use of strong bases or prolonged heating" requires appropriate citation and clarification. The classification of NPhth as base-labile is inaccurate. NPhth is generally stable under basic conditions (until it is boiled in presence of hydrazine) and is not affected during acyl group removal with NaOMe. Therefore, the authors are requested to revise this statement to accurately reflect the stability profile of NPhth group.
5. The statement on Page 1, left column "These conditions can compromise sensitive glycosidic bonds and other protecting groups used in AGA, making them incompatible with the AGA process (Fig. 1A)" is misleading and requires revision. These protecting groups are not typically removed during the automated glycan assembly (AGA) process itself, but rather during the final global deprotection steps after synthesis is complete. Therefore, the concern regarding incompatibility with AGA is not specific to this platform and instead applies broadly to all multistep oligosaccharide syntheses. The authors are requested to revise this statement accordingly for accuracy.
6. The statement on Page 1, right column "The cleavage of multiple TCA and Cbz groups requires prolonged reaction times and often faces selectivity issues" would benefit from supporting references. Please provide specific examples or literature references that demonstrate challenges or instances of selective removal of TCA and Cbz protecting groups from the C2-amine position during the synthesis of HMOs or other oligosaccharides.
7. The sentence on Page 1, right column "Key to this development was the introduction of the p-nitrobenzyloxycarbonyl (pNZ)^(26–28) protecting group that was originally developed for side chain protection in peptide chemistry to mask the amine in several glucosamine building blocks" is bit confusing and should be rephrased for clarity.
8. Since the authors were not the first to introduce the p-nitrobenzyloxycarbonyl (pNZ) group for protecting the amine functionality on sugars, they should acknowledge the original work that first applied pNZ to sugar amines. It would also be appropriate to summarize the key observations from that study, particularly regarding the reactivity of pNZ-protected donors in solution-phase glycosylation compared to donors bearing other common amine protecting groups.
9. Since thioglycoside 1 has been previously reported in doi.org/10.1002/ejoc.200700048, the authors should cite this reference wherever appropriate in the manuscript to ensure proper attribution of the original work.
10. Is there a specific rationale for selecting the STol leaving group over more commonly used thioglycosides such as SET or SPh for HMO assembly via AGA? Additionally, have the authors evaluated whether the other class of leaving group has any impact on the reactivity or performance of donors bearing the C2-pNZ protecting group?
11. Since the authors claim to introduce a new protecting group at the C2-amino position of glucosamine, it would be valuable to evaluate and discuss the glycosylation efficiency of C2-pNZ-protected glucosamine donors in comparison to their C2-NHTroc and C2-Phth counterparts for synthesis of oligosaccharide 10. Such a comparison would help contextualize the advantages or limitations of the pNZ group and further support its utility in automated glycan assembly.
12. The statement on Page 2, right column "Generally, glycosylation efficiency and stereoselective outcomes are influenced by many factors, including concentration, temperature, protecting groups, and solvent." accurately reflects well-established

knowledge in the field. However, glycosylation reactions have been studied for nearly 140 years, and numerous seminal studies have contributed to our understanding and optimization of these influencing factors. It is therefore not appropriate that three out of the four cited references are the authors' own publications unless authors exclusively referring to AGA.

13. The nomenclature used for some glycosyl building blocks in the manuscript is inconsistent. For example, "Glucose 11" should be more accurately referred to as "thioglycoside 11." Similarly, other donors and acceptors should be correctly labeled throughout the manuscript e.g., "galactose XX" or "glucosamine XX" should be revised to reflect their actual chemical form (such as thioglycoside or protected acceptor/donor), where applicable.

14. The sentence on Page 3, right column "Bis glycosylation with glucosamine building block 3 added the terminal galactose at both the 6-OH and 3-OH positions" is unclear and potentially misleading. It gives the impression that the glucosamine donor is adding galactose residues, which are not chemically accurate.

15. On Page 3, right column, the authors state that "pNZ-protected hexasaccharide 25a is more acid stable than TCA-protected hexasaccharide 25b over multiple glycosylation cycles." This statement requires clarification. Are the authors suggesting that the NHTroc group degrades under the glycosylation conditions, or that certain glycosidic linkages are undergoing acid-catalyzed hydrolysis (due to C2-NHTroc) during the synthesis? Additionally, there appears to be a discrepancy in the ESI, where compounds 25a and 25b are labeled as 24a and 24b. Please correct this for consistency.

16. The statement on Page 5, left column "The pNZ group was removed by reduction of the nitro group in a two-step protocol. pNZ was first reduced to the p-aminobenzyloxycarbonyl derivative that underwent a spontaneous 1,6-electron pair shift, affording quinonimine methide and carbamic acid. The transient carbamic acid subsequently decomposed to release the free amine." describes a well-known deprotection mechanism of pNZ-type protecting groups. However, the authors should provide a relevant literature citation to support this mechanistic description.

17 On Page 5, left column, References 23 and 39 do not appear to support the statement regarding the removal of the pNZ group using SnCl₂ and Zinc-copper couple, respectively. Please cross-check these citations and correct them accordingly. The authors are advised to carefully review and verify all citations throughout the manuscript to ensure they accurately support the associated statements throughout the text.

18. The sentence on Page 5, left column "In the second step, N-acetylation converted the free glucosamine to GlcNAc" is misleading as currently phrased. It gives the impression that free glucosamine was used as a starting material, whereas the N-acetylation is actually being performed on glucosamine residues present within oligo-LacNAc or HMO structures. Please revise the sentence for clarity.

19. The statement on Page 5, right column "Three methods for the selective removal of the pNZ group were developed to expand the toolbox for producing synthetic oligosaccharides with the Glycoeer® automated synthesizer" is potentially misleading. It implies that these deprotection methods were developed by the authors for the first time. If these methods have been previously reported or adapted from earlier studies, the authors should clarify this point and provide appropriate citations. The wording should be revised to accurately reflect the novelty and contribution of the present work.

21. On Page 5, the caption for Figure 5B refers to the panel as a "collection of all HMOs." However, the first three structures shown are not HMOs but rather LacNAc-based oligosaccharides. Please correct it.

Version 1:

Reviewer comments:

Reviewer #1

(Remarks to the Author)

I am satisfied with the revisions made and the rebuttals to my comments and those from the other reviewers.

I recommend acceptance of the manuscript for publication.

Reviewer #3

(Remarks to the Author)

The revised manuscript by Professor Seeberger and coworkers has addressed all my queries and incorporated the relevant suggestions. The authors have clearly demonstrated the importance of the p-nitrobenzyloxycarbonyl (pNZ) protecting group at the C2-amino position of glucosamine in enabling the efficient synthesis of HMOs via the automated glycan platform. They have also included all the relevant references regarding the pNZ group. I have no further comments or suggestions and recommend publishing the revised manuscript as it is.

Referee 1

1. Comment: A similar comparative study of the process may also be conducted with the carboxybenzyl (Cbz) protecting group on the glucosamine and reported in the SI.

Response: The AGA synthesis of the LNnH hexasaccharide **25c** from *N*-Cbz-protected building block **17b** was performed. The HPLC profile is provided in the SI (Fig.S3). Using the same automation sequence and condition, only a trace amount (6%) of the desired product **25c** was isolated by NP-HPLC, with the major byproduct the result of incomplete coupling (33%). The glycosylation efficiency with both Cbz-protected **17b** and TCA-protected **17a** is lower when using identical conditions as for *p*NZ-protected building block **3**. The result was added to the revised manuscript.

Manuscript Updates: the following was amended (page 3, right column) “For comparison, the *N*-*p*NZ-protected glucosamine donor **3** was replaced by *N*-TCA-protected **17a** and *N*-Cbz **17b** glucosamine donor to obtain hexasaccharides **25b** and **25c**, respectively. The *p*NZ-protected donor afforded a higher isolated yield (29%) over multiple glycosylation cycles, compared to the TCA-protected (16%) and the Cbz-protected donor (6%). In the case of the TCA-protected donor, the absence of prominent deletion sequences suggested that glycosidic linkages may undergo acid-catalyzed hydrolysis during the synthesis.^[52] In contrast, the use of the Cbz-protected donor resulted in the deletion of sequences due to poorer glycosylation efficiency (Fig. S3).”

2. Comment: I recommend that the words asymmetrical and asymmetrically be replaced with unsymmetrical and unsymmetrically.

Response: Changes made.

Referee 2

Comment: The authors emphasize that the use of *p*-nitrobenzyloxycarbonyl (*p*NZ) group as protective group of amino function was critical to achieving efficient synthesis of the target molecules. However, the reviewer does not share this view.

The authors devoted substantial space to detailing the synthetic procedures, including the preparation of various synthetic building blocks and the optimization of glycosylation reaction conditions. However, these descriptions fail to clearly demonstrate the key role played by *p*NZ group. While it is reasonable to believe that the *p*NZ protection may have contributed to improved synthetic efficiency—such as in the comparison of yields between hexasaccharide **25a** and **25b**, where the yield of **25a** was enhanced by the use of *p*NZ—it remains unclear from the manuscript whether this was truly a determining factor for the success of the reactions.

Response:

We respectfully disagree with the reviewer's assessment and clarify that the value of the *p*-nitrobenzyloxycarbonyl (*p*NZ) group is valuable across the entire AGA workflow: (i) building-block preparation, (ii) solid-phase assembly, (iii) global deprotection, and (iv) end-product customization via orthogonal *p*NZ removal options.

1. Proven platform compatibility (rationale and precedent): *p*NZ side-chain protection is well established in solid-phase peptide synthesis and is removed under mild reductive conditions (*Eur. J. Org. Chem.* **2005**, *2005*, 3031-3039). Motivated by this report, we adopted *p*NZ for glucosamine in solid-phase glycan synthesis and observed analogous robustness.

2. Building-block preparation efficiency: *p*NZ-protected glucosamine enabled solubility-guided purifications, significantly reducing chromatography needs and shortening access to clean building blocks.

3. Matched-pair assembly outcomes: Direct comparisons (hexasaccharides **25a** vs **25b** vs **25c**) show consistently higher isolated yields with *p*NZ.

4. Global deprotection reliability: With TCA-protected building blocks, we encountered incomplete deprotection, fucose cleavage, and solubility problems (*Chem. Sci.* **2019**, *10*, 5634–5640; *Chem. Sci.* **2022**, *13*, 2115–2120), sometimes requiring Birch reduction with poor scalability. In contrast, *p*NZ deprotects cleanly under mild, scalable conditions.

5. Three deprotection methods: A further advantage of *p*NZ as demonstrated in the manuscript is its three complementary deprotection modalities, each compatible with different protecting-group constellations. This orthogonality lets us choose conditions that preserve sensitive motifs and tailor final finishing steps.

In sum, *p*NZ is not a marginal convenience but an enabler of scalable AGA: accelerating building block preparation, improving on-resin yields, ensuring reliable global deprotection, and offering flexible end-product customization via three orthogonal removal pathways.

Manuscript Updates: the following sentence “The end-to-end tractability of *p*NZ-protected oligosaccharides underscores the robustness of this approach, while three orthogonal deprotection pathways offer synthetic versatility for HMO compounds.” was added at the end of abstract.

Another sentence “The high end-to-end efficiency of *p*NZ protection facilitated the synthesis of linear, branched, structurally diverse, and complex HMO variants using the Glyconeer automated synthesizer. Moreover, three methods for the selective removal of the *p*NZ group were applied to enable product customization.” was revised at the end of conclusion.

Referee 3

1. Comment: On page 1 left column, reference 6 does not appear to be the appropriate source to support the corresponding statement. Moreover, reference 6 reports that more than 150

HMOs with well-defined structures have been identified, whereas the authors state that "over 100 of those glycans have been fully characterized" which seems inconsistent with the cited source. Additionally, Reference 7 barely talks about HMOs and is not sufficient to support claims regarding their structural elucidation. The authors are advised to cite more relevant and recent reviews or primary research articles that comprehensively address the structural characterization of HMOs.

Response: We thank the reviewer for pointing out the mismatch between our statement and the cited sources. We agree that our original wording conflated historical and current tallies and that Ref. 7 was not the most appropriate to support structural elucidation claims.

Manuscript Updates: The sentence was replaced with "Comprehensive LC–MS surveys and databases have reported more than 200 HMOs, from which over 150 distinct structures were elucidated across both neutral and sialylated HMOs.^[6-9]"

References 7 was replaced with: *J. Agric. Food Chem.* **2006**, *54*, 7471-7480 (ref. 7), *J. Proteome Res.* **2010**, *9*, 4138–4151. (ref. 8) and *J. Proteome Res.* **2011**, *10*, 856–868 (ref. 9).

2. Comment: The following statement requires appropriate citation: Page 1, left column, "However, understanding the functional roles of HMOs remains limited due to difficulties in obtaining these complex glycans in pure form and sufficient quantities. Since isolating large amounts of pure HMOs from natural sources is challenging, synthetic approaches are essential to explore the biological potential of HMOs."

Response: We agree that this statement requires explicit references.

Manuscript Updates: new references were added: *J. Am. Chem. Soc.* **2021**, *143*, 11277–11290 (ref. 10) and *Nutr. Rev.*, **2016**, *74*, 635-644 (ref. 11)

3. Comment: The following statement on Page 1, left column "Automated glycan assembly (AGA) accelerates the synthetic process of a wide variety of complex oligosaccharides" requires appropriate and balanced citation. While the authors reference three of their own publications, it is important to acknowledge that multiple research groups have developed automated glycan synthesis platforms. Please include additional citations to relevant work from other groups to provide a more comprehensive and unbiased overview of advancements in this area.

Response: We agree.

Manuscript Updates: new references were added: *Org. Lett.*, **2012**, *14*, 3036-3039 (ref. 27), *Org. Lett.* **2015**, *17*, 2642–2645 (ref. 28), and *Nat. Synth.*, **2022**, *1*, 854-863 (ref. 29), were added after this statement.

4. Comment: The following statement on Page 1, left column "However, NHTroc and NPhth are base-labile and require rather harsh deprotection conditions, such as use of strong bases or prolonged heating" requires appropriate citation and clarification. The classification of NPhth as base-labile is inaccurate. NPhth is generally stable under basic conditions (until it is boiled in presence of hydrazine) and is not affected during acyl group removal with NaOMe. Therefore, the authors are requested to revise this statement to accurately reflect the stability profile of NPhth group.

Response: Statement revised and reference added. We clarified that our AGA discussion refers to compatibility constraints arising from deprotection modules, not intrinsic base lability per se.

Manuscript Updates: The sentence was updated “However, the Troc group is base-labile, making it incompatible with AGA protocols where concentrated basic solutions are introduced periodically. In contrast, Phth is base-stable and thus compatible with basic steps, but its removal typically relies on hydrazinolysis or related nucleophiles, which can demand large excess of reagents and, depending on substrate, elevated temperature or prolonged reaction times.^{[30, 31]” Chem. - Asian J., 2012, 7, 2482-2501 (ref. 30) and Liebigs Ann., 1997, 1997, 791-802 (ref. 31)}

5. Comment: The statement on Page 1, left column “These conditions can compromise sensitive glycosidic bonds and other protecting groups used in AGA, making them incompatible with the AGA process (Fig. 1A)” is misleading and requires revision. These protecting groups are not typically removed during the automated glycan assembly (AGA) process itself, but rather during the final global deprotection steps after synthesis is complete. Therefore, the concern regarding incompatibility with AGA is not specific to this platform and instead applies broadly to all multistep oligosaccharide syntheses. The authors are requested to revise this statement accordingly for accuracy.

Response: Statement revised.

Manuscript Updates: “These conditions can compromise sensitive glycosidic bonds and other temporary protecting groups, making them not applicable to further modifications or global deprotections following AGA (Fig. 1A).”

6. Comment: The statement on Page 1, right column “The cleavage of multiple TCA and Cbz groups requires prolonged reaction times and often faces selectivity issues” would benefit from supporting references. Please provide specific examples or literature references that demonstrate challenges or instances of selective removal of TCA and Cbz protecting groups from the C2-amine position during the synthesis of HMOs or other oligosaccharides.

Response: We now (i) cite primary examples where TCA/Cbz removal at C2-amine created bottlenecks and (ii) clarify the selectivity issue when Cbz is present both on GlcN (C2) and on a Cbz-bearing linker.

Our prior work (*Chem. Sci.* **2019**, *10*, 5634–5640) showed that global deprotection of Lewis antigens (including Le^x dimer and KH-1) containing multiple TCA-protected glucosamines failed due to fucose cleavage, incomplete TCA reduction, and severe solubility problems; only after switching to Birch conditions, a workaround with practical and scalability limitations, were the final targets obtained. Delbianco et al. similarly reported analogous difficulties (trace products; *Chem. Sci.* **2022**, *13*, 2115–2120), underscoring that multiple TCA/Cbz removals can require prolonged conditions and are context-sensitive.

Cbz selectivity problem when also present on the linker. In standard AGA workflows using MeNV-type photolabile linkers, Pd/C hydrogenolysis removes TCA, and the Cbz on the amino linker in one step, furnishing multiple free amines. This complicates selective N-acetylation at glucosamine, exemplifying a selectivity issue intrinsic to concurrent Cbz removals.

Manuscript Updates: “Simultaneous cleavage of multiple TCA and Cbz protecting groups often requires extended hydrogenolysis and can suffer from selectivity issues, particularly when Cbz is also present on linkers, leading to parallel unmasking of multiple amines and complicating selective N-acetylation. Prior experiments reported incomplete TCA reduction, fucose cleavage, and solubility problems.^{[33, 34]”}

7. Comment: The sentence on Page 1, right column “Key to this development was the introduction of the *p*-nitrobenzyloxycarbonyl (*p*NZ)^(26–28) protecting group that was originally

developed for side chain protection in peptide chemistry to mask the amine in several glucosamine building blocks” is bit confusing and should be rephrased for clarity.

Response: Changes made.

Manuscript Updates: “A Key aspect of this development was the introduction of the *p*-nitrobenzyloxycarbonyl (*p*NZ)^[35-37], a carbamate-type protecting group. Initially, *p*NZ was employed in peptide chemistry for the protection of amino side chains, particularly to mask the amine functionalities of several amino acids.”

8. Comment: Since the authors were not the first to introduce the *p*-nitrobenzyloxycarbonyl (*p*NZ) group for protecting the amine functionality on sugars, they should acknowledge the original work that first applied *p*NZ to sugar amines. It would also be appropriate to summarize the key observations from that study, particularly regarding the reactivity of *p*NZ-protected donors in solution-phase glycosylation compared to donors bearing other common amine protecting groups.

Response: Statement revised and references added

Manuscript Updates: “In these solution-phase studies,^[36, 38] *p*NZ-protected donors exhibit lower reactivity compared to other amino-protected donors, necessitating higher temperatures and longer reaction times. However, in solid phase synthesis, this moderate reactivity helps suppress donor decomposition and side reactions, thereby improving synthetic efficiency.^[24, 39, 40]”

References cited: *Eur. J. Org. Chem.* **2007**, 3392–3401 (ref. 36), *Carbohydr. Res.* **1990**, *202*, 151-164 (ref. 38), *J. Am. Chem. Soc.* **2024**, *146*, 18320–18330 (ref. 24), *Can. J. Chem.* **2002**, *80*, 889-893 (ref. 39) and *Angew. Chem. Int. Ed.* **2022**, *61*, e202115433 (ref. 40) were cited to support this statement.

9. Comment: Since thioglycoside **1** has been previously reported in doi.org/10.1002/ejoc.200700048, the authors should cite this reference wherever appropriate in the manuscript to ensure proper attribution of the original work.

Response: This original work that introduces the *p*-nitrobenzyloxycarbonyl (*p*NZ) group, is already cited in the original manuscript. For clarity and completeness, it has now been cited again following the mention of thioglycoside **1**.

Manuscript Updates: *Eur. J. Org. Chem.* **2007**, 3392–3401 (ref. 36) is cited again after sentence “Thioglycoside **1** containing a C-2 *p*NZ group was synthesized from D-glucosamine in four steps”.

10. Comment: Is there a specific rationale for selecting the STol leaving group over more commonly used thioglycosides such as SEt or SPh for HMO assembly via AGA? Additionally, have the authors evaluated whether the other class of leaving group has any impact on the reactivity or performance of donors bearing the C2-*p*NZ protecting group?

Response: We selected S-*p*-tolyl (STol) thioglycosides for AGA because they offer a practical balance of reactivity, robustness, and handling:

1. Operational handling & purification: STol donors and intermediates are frequently crystalline/solid and easier to purify than their SEt analogues; HSTol is a non-volatile solid, which simplifies weighing and avoids the strong odor/toxicity issues common to volatile SEt/SPh thiols, reducing reliance on extensive chromatography.

2. Balanced activation window: Reactivity scales place SEt > STol > SPh for donors with identical protecting patterns (*Can. J. Chem.* **2002**, *80*, 889–893; *Angew. Chem. Int. Ed.* **2022**,

61, e202115433). In practice, SET can over-activate, correlating with faster donor decomposition/side reactions under catalytic activation (*J. Am. Chem. Soc.* **2024**, *146*, 18320–18330), whereas SPH often requires hotter/longer activation. STol sits in the middle, readily activated under standard AGA promoters (e.g., NIS/TfOH, DMTST) yet resistant to background degradation, which is advantageous on solid support with repetitive cycles.

3. Impact of leaving group with C2-*p*NZ donors: As noted in our manuscript, C2-*p*NZ protection tends to lower donor reactivity relative to classical *N*-acyl protections in solution-phase studies, necessitating higher temperatures/longer times. In the AGA setting, pairing *p*NZ at C2 with STol provides a complementary match: the donor's moderate intrinsic reactivity is not overdriven (mitigating decomposition), while STol's mid-range leaving ability still allows clean activation at –20 to 0 °C with standard promoters. This combination reduced off-cycle consumption and helped maintain high on-resin conversions in our sequences.

11. Comment: Since the authors claim to introduce a new protecting group at the C2-amino position of glucosamine, it would be valuable to evaluate and discuss the glycosylation efficiency of C2-*p*NZ-protected glucosamine donors in comparison to their C2-NHTroc and C2-Phth counterparts for synthesis of oligosaccharide 10. Such a comparison would help contextualize the advantages or limitations of the *p*NZ group and further support its utility in automated glycan assembly.

Response: As detailed in 4th comment above, in the context of AGA workflow, both NHTroc and NPhth are not compatible in practical use and were not re-evaluated again in this study. However, we investigated the use of the carboxybenzyl (Cbz) group at the C2 position of the glucosamine and synthesized the corresponding hexasaccharide **25c**, analogous to compounds **25a** and **25b**. The HPLC profile is provided in the SI (Fig.S3). Following the same synthetic protocol as for the *p*NZ-protected glucosamine, only 6% of the desired product **25c** was isolated by NP-HPLC. The HPLC analysis revealed substantial formation of the deletion sequence (33%), indicating low glycosylation efficiency when employing Cbz-protected glucosamine donor. The result for the Cbz-protected compound has been included in the manuscript, and the corresponding description was added.

On the other hand, Relative Reactivity Value (RRV) experiments were performed to compare the reactivity of TCA-protected and *p*NZ-protected glucosamine donors. Following the established protocol (*Nat. Protoc.* **2007**, *1*, 3143-3152), TCA-protected donor was used as the reference donor, with its RRV value (458) previously reported (*Chem. Eur. J.* **2024**, *30*,

e202400479), while the *p*NZ-protected donor was an unknown donor. The experimental results revealed that the reactivity of TCA-protected donor is 1.1 times higher than of *p*NZ-protected donor, corresponding to an RRV of 415 for 4-O-Fmoc *p*NZ-protected donor. This (unpublished) finding indicates that the reactivities of 4-O-Fmoc TCA-protected and 4-O-Fmoc *p*NZ-protected donors are relatively similar, both exhibiting favorable reactivity.

Manuscript Updates: The following was amended (page 3, right column) “For comparison, the *N*-*p*NZ-protected glucosamine donor **3** was replaced by *N*-TCA-protected **17a** and *N*-Cbz **17b** glucosamine donor to obtain hexasaccharides **25b** and **25c**, respectively. The *p*NZ-protected donor afforded a higher isolated yield (29%) over multiple glycosylation cycles, compared to the TCA-protected (16%) and the Cbz-protected donor (6%). In the case of the TCA-protected donor, the absence of prominent deletion sequences suggested that glycosidic linkages may undergo acid-catalyzed hydrolysis during the synthesis.^[52] In contrast, the use of the Cbz-protected donor resulted in the deletion of sequences due to poorer glycosylation efficiency (Fig. S3).”

12. Comment: The statement on Page 2, right column “Generally, glycosylation efficiency and stereoselective outcomes are influenced by many factors, including concentration, temperature, protecting groups, and solvent.” accurately reflects well-established knowledge in the field. However, glycosylation reactions have been studied for nearly 140 years, and numerous seminal studies have contributed to our understanding and optimization of these influencing factors. It is therefore not appropriate that three out of the four cited references are the authors' own publications unless authors exclusively referring to AGA.

Response: more references added

Manuscript Updates: *Eur. J. Org. Chem.* **2021**, 3251–3259 (ref. 46), *ACS Cent. Sci.* **2021**, 7, 1454–1462 (ref. 47) and doi.org/10.1002/9783527697014.ch1 (ref. 48) were added after this statement.

13. Comment: The nomenclature used for some glycosyl building blocks in the manuscript is inconsistent. For example, “Glucose 11” should be more accurately referred to as “thioglycoside 11.” Similarly, other donors and acceptors should be correctly labeled throughout the manuscript e.g., “galactose XX” or “glucosamine XX” should be revised to reflect their actual chemical form (such as thioglycoside or protected acceptor/donor), where applicable.

Response: Changes made.

14. Comment: The sentence on Page 3, right column “Bis glycosylation with glucosamine building block **3** added the terminal galactose at both the 6-OH and 3-OH positions” is unclear and potentially misleading. It gives the impression that the glucosamine donor is adding galactose residues, which are not chemically accurate.

Response: Changes made.

Manuscript Updates: The statement was changed to “Bis-glycosylation using 4-Fmoc glucosamine donor **3** was performed at both the 6-OH and 3-OH positions of the galactose unit.”

15. Comment: On Page 3, right column, the authors state that “*p*NZ-protected hexasaccharide **25a** is more acid stable than TCA-protected hexasaccharide **25b** over multiple glycosylation cycles.” This statement requires clarification. Are the authors suggesting that the NHTroc group degrades under the glycosylation conditions, or that certain glycosidic linkages are undergoing acid-catalyzed hydrolysis (due to C2-NHTroc) during the synthesis?

Additionally, there appears to be a discrepancy in the ESI, where compounds **25a** and **25b** are labeled as **24a** and **24b**. Please correct this for consistency.

Response: Changes made for compound number labelling in the ESI (Fig.S3). Additionally, we believe the reviewer is inquiring about the NHTCA group degradation, rather than the NHTroc. So, the following statement revision will focus on the NHTCA group described. In this literature (*J. Org. Chem.* **2016**, *81*, 5866–5877), the overall yield of glycan is clearly dependent on the acid concentration and the reaction temperature during the automated assembly synthesis (Table 1).

Manuscript Updates: “In the case of the TCA-protected donor, the absence of prominent deletion sequences suggested that glycosidic linkages may undergo acid-catalyzed hydrolysis during the synthesis.^{[52]” (*J. Org. Chem.* **2016**, *81*, 5866–5877) (ref. 52)}

16. Comment: The statement on Page 5, left column “The *p*NZ group was removed by reduction of the nitro group in a two-step protocol. *p*NZ was first reduced to the *p*-aminobenzoyloxycarbonyl derivative that underwent a spontaneous 1,6-electron pair shift, affording quinonimine methide and carbamic acid. The transient carbamic acid subsequently decomposed to release the free amine.” describes a well-known deprotection mechanism of *p*NZ-type protecting groups. However, the authors should provide a relevant literature citation to support this mechanistic description.

Response: Since the same reference also described the catalyst amount of acid that can reduce reaction time, we cited it after the sentence, “The transformation was accelerated in the presence of an acid catalyst” (following sentence) to avoid frequently repeated referencing. However, to ensure clarity and prevent any misunderstanding, the same reference has been cited again after the mechanism description.

Manuscript Updates: *Eur. J. Org. Chem.* **2005**, 3031–3039 (ref. 35) was cited one more time after this statement.

17. Comment: On Page 5, left column, References 23 and 39 do not appear to support the statement regarding the removal of the *p*NZ group using SnCl₂ and Zinc–copper couple, respectively. Please cross-check these citations and correct them accordingly. The authors are advised to carefully review and verify all citations throughout the manuscript to ensure they accurately support the associated statements throughout the text.

Response: The SnCl₂ deprotection of the *p*NZ group is presented in Table 2 of *Eur. J. Org. Chem.* **2005**, 3031–3039. The Zinc-copper reduction of a nitro group, using 4-nitrostilbene as a model substrate, is illustrated in Figure 1 of *Carcinogenesis* **1986**, *7*, 183-184. In this study, we slightly modified the conditions (*Carcinogenesis* **1986**, *7*, 183-184, and *Eur. J. Org. Chem.* **2005**, 3031–3039) to apply the removal of the *p*NZ group.

Manuscript Updates: The references were replaced with *Eur. J. Org. Chem.* **2005**, 3031–3039 (ref. 35), and *Carcinogenesis* **1986**, *7*, 183-184 (ref. 54).

18. Comment: The sentence on Page 5, left column “In the second step, *N*-acetylation converted the free glucosamine to GlcNAc” is misleading as currently phrased. It gives the impression that free glucosamine was used as a starting material, whereas the *N*-acetylation is actually being performed on glucosamine residues present within oligo-LacNAc or HMO structures. Please revise the sentence for clarity.

Response: Changes made.

Manuscript Updates: The sentence was revised to read “In the second step, *N*-acetylation was conducted to further modify the amino group of glucosamine residues in oligo-LacNAc and HMO structures, resulting in the formation of *N*-acetylglucosamine within these structures.”

19. Comment: The statement on Page 5, right column “Three methods for the selective removal of the *p*NZ group were developed to expand the toolbox for producing synthetic oligosaccharides with the Glyconeer® automated synthesizer” is potentially misleading. It implies that these deprotection methods were developed by the authors for the first time. If these methods have been previously reported or adapted from earlier studies, the authors should clarify this point and provide appropriate citations. The wording should be revised to accurately reflect the novelty and contribution of the present work.

Response: Three methods already published in *Carcinogenesis* **1986**, *7*, 183-184, *Eur. J. Org. Chem.* **2005**, 3031–3039 and *J. Org. Chem.* **2022**, *87*, 910–919 (already cited). However, for the B₂(OH)₄/4,4'-bipyridine condition, the authors demonstrated the reduction of the nitro group with 3-nitrostyrene substrate. In this study, we slightly modified the condition and applied it to the carbohydrate substrate for the first time. To avoid the misunderstanding, these reported will be cited in this statement.

Manuscript Updates: The statement was changed to read “The high end-to-end efficiency of *p*NZ protection facilitated the synthesis of linear, branched, structurally diverse, and complex HMO variants using the Glyconeer automated synthesizer. Moreover, three methods for the selective removal of the *p*NZ group were applied to enable product customization”.

20. Comment: On Page 5, the caption for Figure 5B refers to the panel as a “collection of all HMOs.” However, the first three structures shown are not HMOs but rather LacNAc-based oligosaccharides. Please correct it.

Response: Changes made.

Manuscript Updates: The caption for Figure 5B was changed to “Collection of neutral linear, symmetrical- and unsymmetrical-branched LacNAc-based oligosaccharides and HMOs”